# Biomimetic Polymer-Based Engineered Scaffolds for Improved Stem Cell Function

**DOI:** 10.3390/ma12182950

**Published:** 2019-09-11

**Authors:** Dinesh K. Patel, Ki-Taek Lim

**Affiliations:** 1The Institute of Forest Science, Kangwon National University, Chuncheon-24341, Korea; dineshbhud10@gmail.com; 2Department of Biosystems Engineering, College of Agriculture and Life Sciences, Kangwon National University, Chuncheon-24341, Korea

**Keywords:** scaffolds, extracellular matrices, cell functions, stem cells, tissue engineering

## Abstract

Scaffolds are considered promising materials for tissue engineering applications due to their unique physiochemical properties. The high porosity and adequate mechanical properties of the scaffolds facilitate greater cell adhesion, proliferation, and differentiation. Stem cells are frequently applied in tissue engineering applications due to their excellent potential. It has been noted that cell functions are profoundly affected by the nature of the extracellular matrix (ECM). Naturally derived ECM contains the bioactive motif that also influences the immune response of the organism. The properties of polymer scaffolds mean they can resemble the native ECM and can regulate cellular responses. Various techniques such as electrospinning and 3D printing, among others, are frequently used to fabricate polymer scaffolds, and their cellular responses are different with each technique. Furthermore, enhanced cell viability, as well as the differentiation ability of stem cells on the surface of scaffolds, opens a fascinating approach to the formation of ECM-like environments for tissue engineering applications.

## 1. Introduction 

Stem cells have received a great amount of attention from academia as well as industry for tissue engineering and regenerative medicine applications due to their exclusive differentiation potential and pluripotency [1,2,3,4]. Pluripotent stem cells (PSCs) have the ability to self-renew and they can generate all the types of cell in the body. Embryonic stem cells (ESCs) and induced pluripotent stem cells (iPSCs) are the types of PSCs which are utilized in the field of tissue engineering. The potential of these types of stem cells is governed by their surrounding microenvironments [5]. Several factors such as hormones, growth factors, chemical moieties, and the extracellular matrix (ECM) conditions have wide influence on the differentiation and pluripotency of stem cells [6,7,8,9,10,11]. Figure 1 represents the different interactions of stem cells with their surrounding microenvironments, which determine their fate [5]. 

The ECM provides the required physical, chemical, and mechanical support to the cells for better growth and differentiation [12]. The interactions between ECM and cells are extremely affected by the surface morphology of the ECM [13]. These materials have drawn a significant amount of interest regarding cellular activities owing to their similarity in intrinsic morphology with native ECM [14]. Notably, better cell proliferation and differentiation were observed on the surface of scaffolds due to their porosity and adequate mechanical properties which facilitate the easy transport of gases, nutrients, and other regulatory factors [15]. The chemical composition, surface chemistry, porosity, degradation behavior, and mechanical strength of an ECM play very important roles in determining stem cell fate [16]. These materials can be biological or synthetic and degradable or nondegradable in nature [17]. The biological scaffolds are derived from living sources, such as human and animal tissues, whereas synthetic scaffolds are prepared from various biocompatible polymers [18]. It is well known that for tissue engineering applications, materials should be biocompatible and biodegradable in nature. Additionally, materials should also have adequate mechanical strength to support cellular activities [19]. The synthetic biodegradable polymer scaffolds have an additional benefit over nondegradable materials—they do not require extra attention to be removed from the biological conditions. Several biodegradable synthetic polymers, such as polycaprolactone (PCL), poly-l-lactic acid (PLA), and polybutylene terephthalate (PBT), among others, are frequently used to fabricate porous scaffold materials for tissue engineering applications [20,21,22]. The biodegradability of the scaffold materials can be tuned by using a suitable filler/additive in their matrices or blending with other polymers. Different metals and their oxides and carbon in various forms, such as graphene, carbon nanotubes (CNTs), fullerene, clay or modified clay, and zeolites, are intensively used to alter the properties of the pure polymers [23,24,25,26]. These nanomaterials have different kinds of morphology as well as orientation (1D, 2D, and 3D) that considerably influence the interactions between the polymer and nanomaterials, and consequently, cellular activities [27]. Although there are several reports available where scaffold materials have been utilized in the field of tissue engineering for medicine, surgery, and dental applications, their results exhibited that rapid regeneration of the damaged tissues, namely enamel, pulpodentin complex, periodontal apparatus, and teeth, occurred in the presence of the scaffold materials. It has been seen that the hard scaffolds are suitable for bone tissues, whereas soft and injectable scaffolds are more useful for pulpodentin and periodontal complexes. Stem cells in combination with scaffolds and signaling molecules, have been already developed for the regeneration of damaged dental tissues. However, several concerns, such as the safety and standardization of the techniques, still have to be resolved for clinical application in humans [28,29]. Herein, we have briefly described the major classes of the biodegradable polymer scaffolds and their effects on stem cell functions. We have also mentioned some common techniques of scaffold fabrication and their application. Moreover, we have endeavored to explain the effects of the different types of naturally derived ECM on immune responses. This review is focused on the different kinds of polymer scaffolds and their tissue engineering applications using stem cells.

## 2. Types of Polymer Scaffolds

The scaffold materials are extensively used in tissue engineering fields due to their superior physiochemical properties. The physiochemical behaviors such as external geometry, surface properties, degradable nature, porosity and pore size, mechanical strength, biodegradability, and biocompatibility of the scaffolds play very important roles in cell adhesion and proliferation [30]. Figure 2 shows the flowchart for the generation of active tissues from biomaterials [31]. Here, we have described some common types of scaffolds and their advantages for tissue engineering.

### 2.1. Three-Dimensional (3D) Polymer Scaffolds 

Three-dimensional (3D) printing techniques have received a great amount of attention in the field of tissue engineering owing to their ability to design and fabricate complex structures with high accuracy. In this process, the materials are deposited in a layer-by-layer fashion to form objects from 3D model data [32]. This is an attractive technique due to the high levels of build resolution, smooth surfaces, rapid building, good mechanical strength between the layers, and efficacy in printing clear objects, which are not achievable with the conventional 2D model [33]. It has been noted that the cell responses were different in the 3D microenvironments compared to those of 2D models [34]. The porous 3D scaffolds facilitate better cell–cell and cell–matrix interactions and lead to a significant enhancement of the cell densities compared to the 2D conditions [35]. Notably, better growth of blood vessels with increased nutrient, oxygen, and waste diffusion is possible in the 3D microenvironments [36,37,38]. Polylactic acid (PLA); polycarbonate (PC); acrylonitrile butadiene styrene; oligo-propylene fumarates; pluronic, alginate, gelatin, and hyaluronic acid; epoxy resins, etc. are frequently used in 3D-printing techniques for different applications [39,40,41]. The 3D-printed polymer materials can be used in the aerospace industry for the manufacturing of lightweight complex structures, the architectural industry for structural models, and also in artistic disciplines for artifact replication [42,43]. However, the weak mechanical strength of the 3D-printed polymer materials restricts it to use in load-bearing functional applications. The mechanical strength of printed materials can be improved by using a suitable filler in their matrix. 

### 2.2. Hydrogel Scaffolds

Hydrogel scaffolds have acquired significant attention from researchers for their use as biomaterials for tissue engineering applications. The hydrogel can enable molecularly tuned biofunctions and adequate mechanical strength, as well as ECM-like conditions for better cellular activity. Hydrogel scaffold is the kind of structural support made by the bioactive compounds. Hydrogels are composed of the crosslinked networks of the polymer chains that are formed in the presence of water or a physiological medium [44]. This network structure gives the hydrogels a unique swelling potential and 3D skeleton. This network structure can be formed through the chemical interactions between the different functional groups present in the matrix or through physical interactions [45,46,47]. Depending on this structure, hydrogels may be chemically stable or unstable in nature, which influences the solubility of hydrogels in various solvents. The unstable gels are known as a reversible or physical gel and they are inhomogeneous in nature [48]. Different synthetic and natural polymers are often used for hydrogel synthesis. Hydrogels synthesized from the common synthetic and natural polymers are represented in Table 1 [49]. Naturally derived hydrogels have several advantages over synthetic polymer scaffolds in terms of their biocompatibility, cellular interactions, and degradation [49]. However, owing to their weak mechanical strength, hydrogel applications are restricted to limited fields. This drawback can be resolved for synthetic polymer-based hydrogels by controlling the structure or functional moieties of the polymer chains or by incorporating appropriate fillers/additives [15]. Biocompatible hydrogels are widely utilized in tissue engineering applications for wound healing, bone regeneration, drug transport, etc. [50]. Besides, hydrogels facilitate the development and differentiation of cells in newly developed tissues in the presence of growth factors [51]. Hydrogels also enable rapid nutrient exchange, cell migration, and angiogenesis [52]. 

### 2.3. Fibrous and Porous Scaffolds

Fibrous scaffolds are another important category of scaffolds and have gathered a lot of interest in tissue engineering applications for neural tissue engineering, bone, cartilage, skeletal muscle, and controlled delivery of small molecules such as drugs, DNA, and proteins [53]. The high aspect ratio of fibrous scaffolds with their intrinsic structure confers better cell adhesion, proliferation, migration, and differentiation [54,55]. Electrospinning, self-assembly, and phase separation techniques are used to fabricate the fibrous scaffolds. Among them, electrospinning is the most versatile technique used for the fabrication of nanofibers [56]. Various synthetic and natural polymers, such as PLA, polyurethane (PU), PCL, poly(lactic-*co*-glycolic acid) (PLGA), and collagen, among others [57,58,59,60,61,62], are commonly employed for nanofiber fabrication for different applications. The properties of fabricated nanofibers can be tuned by changing the functional groups of the pure polymer or blending with other materials. Fabrication of the small molecule-loaded nanofibers can be accomplished by direct mixing of the targeted molecules into the polymer solution [63]. Porous scaffolds, such as foam, with homogenous interconnected pore networks are also considered to be useful ECM platforms for biomedical applications [20]. The porous structure of the scaffold has an auxiliary advantage in cell adhesion and proliferation processes. This structure facilitates proper nutrient transport within the network structure and limits the cluster size to the pore size of the scaffolds. Moreover, the pore architectures can be optimized by exploiting a suitable solvent and phase separating conditions [64]. PLGA, PCL, poly(d,l-lactic acid) (PDLLA), and PBT are some of the synthetic polymers commonly applied for the fabrication of porous scaffolds [21,22,65]. Table 2 represents several common scaffold fabrication techniques and their applications in different fields. For naturally derived ECM, top-down as well as bottom-up approaches have been utilized, including ECM proteins and immunomodulatory domains, such as matrix metalloproteinase (MMP)-sensitive peptides, respectively. 

## 3. Impact of Scaffolds on Stem Cell Functions

Scaffolds prepared from different techniques have different physiochemical properties, ranging from porous to fibrous, irregular to regular, and nanofibrous to microfibrous architectures, which have a great impact on stem cell functions, since the different structure, porosity, and surface roughness of each fabricated scaffold has a direct influence on cell growth and tissue regeneration. In this section, we have described the stem cell fate in the presence of different kinds of scaffolds by considering some common examples with good results. The surface topography from the micro- to nanoscale has a great influence on cell behavior, which can induce the direct osteogenesis of stem cells [89,90,91]. Kumar et al. fabricated PCL-based scaffolds with various structures, such as salt-leached scaffolds, gas-foamed scaffolds, gas-foamed phase-separated scaffolds, nanofiber scaffolds, and spin-coated films, and evaluated their topographical effects on stem cell fate. It was observed that only the nanofibrous scaffolds had the potential to drive human bone marrow stromal cells (hBMSCs) towards osteogenesis without any additional osteogenic supplements. Furthermore, the cell shape was also different on these scaffolds’ surfaces. More elongated and highly branched cell morphologies were observed on nanofibrous surfaces in the absence of osteogenic supplements, whereas rounded and less branched morphologies appeared on the flat surface of the scaffolds in the absence of osteogenic supplements. This was due to the greater adhesion of hBMSCs to the nanofibrous surface that facilitates the osteogenesis and elongated, branched morphology. Figure 3 represents the cell morphologies cultured on different scaffold surfaces. These results indicate that cell behavior is highly influenced by the surface topography of the fabricated scaffolds and the scaffolds’ potential can be altered by changing the scaffold structure [92]. 

Altering the properties of the 3D scaffolds to improve tissue regeneration is a fundamental task for tissue engineering applications. It is well known that the fabrication of functionalized scaffold devices with biomolecules or with growth factors is expensive, and their life span is also limited [93]. Therefore, development of a stable, inexpensive scaffold device is an interesting goal for rapid tissue regeneration. It was noted that the open pore structure of freeform fabricated (FFF) scaffolds supported the rapid proliferation of hBMSCs, but limited osteogenic differentiation. However, electrospun nanofibers have the opposite effect. A combination of FFF and nanofiber scaffold properties will provide a suitable platform for the rapid proliferation as well as differentiation of stem cells. In another study, Kumar and coworkers fabricated FFF scaffolds of poly(ε-caprolactone) (PCL) polymer with roughened struts and evaluated their effects on stem cell proliferation and differentiation in a controlled fashion. They increased the surface roughness of developed FFF scaffolds through a solvent etching technique. It was interesting to note that the etched scaffolds appeared to induce the osteogenic differentiation of stem cells to a greater extent compared to the unetched scaffolds. Moreover, the proliferation potential was the same in both scaffolds. The cells were highly spread on the unetched scaffold’s surface, while more rounded morphology was observed on the etched scaffold’s surface. This indicates that stem cell fate is deeply affected by the roughness of the scaffolds [93]. Since it is well established that stem cells have the potential to differentiate into other types of cell, this potential can be utilized in the field of tissue engineering to solve the shortage of bone and cartilage tissues. Scaffolds play crucial roles in the adhesion, proliferation, and differentiation of stem cells. The chemical composition and physical structure are the important factors of the scaffolds, which suggest their range of applications. Various metals have excellent mechanical strength, which makes them suitable materials for tissue engineering. However, the lack of degradability under biological conditions restricts their wide applicability. Moreover, some ceramic materials such as calcium phosphate, hydroxyapatite, etc. possess superior differentiation potential, but due to poor mechanical strength, their applications are also limited. In contrast, the properties of polymers can be easily modified and they can be used for the fabrication of biomimetic scaffolds for tissue regeneration [94].

The most important challenge in the field of tissue engineering is the fabrication of scaffold devices that have adequate mechanical strength while helping to speed up the recovery of defective bone tissues. The effects of the highly porous 3D nanofibrous poly(l-lactic acid) (PLLA) scaffolds fabricated by a phase separation technique on hMSCs have been studied by Hu et al. Histological analysis and calcium content quantification results revealed that differentiation of hMSCs occurred on the nanofibrous scaffold surface. Detailed study indicated that nanofibrous scaffolds facilitated the development of both bone and cartilage tissues and had the potential for osteochondral construction [95]. It has been noted that the hard tissues such as bone and dentine have a rich source of collagen type I fibrous protein, which consists of ~80–90% organic moieties and is frequently utilized as an ECM for these cells [96]. Pathogen transmission, immune response, and weak mechanical properties restrict its use as an ideal material for tissue engineering. These concerns can be resolved by using the synthetic polymer scaffolds which have suitable mechanical strength, flexible structures, and controlled degradation [97,98]. Dental pulp stem cells (DPSCs) have been considered a suitable cell source for dental tissue regeneration due to their ability to self-renew and excellent proliferation as well as differentiation potential [99,100]. A comparative study was done to evaluate the fate of DPSCs in the presence of nanofibrous (NF) and solid-walled (SW) poly(l-lactic acid) (PLLA) scaffolds. Notably, better cell adhesion, proliferation, and differentiation of DPSCs were reported on the NF scaffolds compared to those of the SW scaffolds. Significant enhancement in alkaline phosphatase (ALP) activity and odontogenic-related genes was observed in the presence of NF scaffolds compared to those of the SW scaffolds. Additionally, higher mineralization was reported in the NF scaffold condition. Furthermore, enhanced odontogenic differentiation and hard tissue formations were observed on the NF scaffolds compared to the SW scaffolds in nude mice after 8 weeks of transplantation [101]. This is attributed to the biomimetic porous structure of the scaffolds that provides the natural ECM microenvironment to the dentine tissue for proper cell attachment, proliferation, and tissue neogenesis [93]. Figure 4 represents the growth of DPSCs on NF and SW PLLA scaffolds at different time intervals. This result clearly indicates the presence of greater cell densities on NF scaffolds than SW scaffolds. NF scaffolds conferred better ECM conditions for cell attachment and differentiation compared to SW scaffolds [101]. 

Chitosan, a natural biomaterial, is employed as an ECM platform for tissue engineering applications due to its superior physiochemical properties and excellent biocompatibility. It has been noted that induced chondrogenesis was observed in the presence of pure chitosan scaffolds. Ragetly et al. prepared type II collagen-coated chitosan fibrous scaffolds through a wet- spinning process and evaluated their effects on mesenchymal stem cells (MSCs) in terms of cell adhesion and chondrogenesis. The cell attachment is widely affected by the presence of type II collagen. Type I collagen has been more extensively applied for tissue engineering applications than type II collagen, so the utilization of type II collagen is more fascinating for the development of biomimetic scaffolds in the field of tissue engineering. An enhancement in cell adhesion as well as chondrogenesis was observed in the presence of the developed biomimetic scaffolds compared to the pure chitosan scaffolds. This improvement was due to the presence of type II collagen in biomimetic scaffolds, which facilitated the better cell attachment through interactions with different binding sites. Furthermore, an enhanced cell distribution was also observed in type II collagen- coated chitosan scaffolds. An improved amount and quality of the ECM was noted in type II collagen-coated scaffolds after 21 days of treatment compared to the pure chitosan scaffolds. This enhancement in ECM content is responsible for better cellular activity [102]. Biomaterial scaffolds play a critical role in the regulation of cell growth and differentiation through their mechanical support as well as geometry. Native ECM contains attractive materials for scaffold fabrication, such as alginate, collagen, etc., because of their specific cell proliferation and differentiation abilities [103,104,105]. Rowland and coworkers evaluated the effects of the different crosslinking agents on the chondrogenic differentiation of MSCs in the presence of cartilage-derived matrix (CDM) scaffolds. It was observed that the scaffolds developed from CDM conferred a chondroinductive microenvironment that facilitated cartilaginous matrix synthesis in the cells. However, the tendency of CDM to shrink in the cultured media restricts its potential broad application. The authors applied dehydrothermal (DHT) and ultraviolet light (UV) treatment and the chemical crosslinker carbodiimide (CAR) on the CDM scaffolds. It was interesting to note that the original shape of the CDM scaffolds was maintained despite these treatments. DHT and UV treatments facilitate cell attachment, while CAR treatment inhibits cell adhesion. Furthermore, the chondrogenic differentiation of MSCs is prominently affected by these treatments [106]. Cellular behavior is also affected by the surface properties such as the wettability, roughness, etc. of polymers, and this can be altered by changing the chemical components or by incorporating suitable fillers. Poly(caprolactone) (PCL) has gained a considerable amount of attention for use as a 3D-printed material for tissue engineering applications owing to its superior mechanical strength and biocompatibility [107,108]. Despite this promising potential, PCL is hydrophobic in nature, which may limit its range of applications. Rashad and coworkers fabricated PCL/cellulose nanofibril (CNF) 3D-printed scaffolds and evaluated the cellular responses using MSCs. The hydrophobic nature of the PCL polymer was tuned by incorporating the desired amount of multiscale hydrophilic and biocompatible CNF filler into the matrix. The significant enhancement in the hydrophilic properties of PCL scaffolds was observed in the presence of CNF. Live/dead staining and dehydrogenase release assay indicated that CNFs have no adverse effects on MSCs. An enhancement in cell viability, attachment, proliferation, and osteogenic differentiation of MSCs occurred in the presence of the PCL/CNF-based 3D-printed scaffolds compared to the pure PCL scaffolds. Furthermore, enhanced alkaline phosphatase (ALP) activity and collagen type I and mineral formations were observed in CNF-coated scaffolds. This enhancement was due to the wrapping of the hydrophilic CNFs by hydrophobic PCL layers, which provides a suitable platform for enhanced cellular activities. Figure 5 represents CNF-coated 3D scaffolds of PCL and their effects on MSC fate [109]. 

These results clearly indicate that the collagen type I production was higher in CNF-coated conditions than in the control. The proliferation and differentiation potential of hMSCs are highly governed by the material’s nature, signaling patterns, and other external factors such as the magnetic and electric fields [110,111,112,113]. Hence, it is necessary to prepare the scaffolds with proper features that provide a better microenvironment for MSC growth. Jin et al. fabricated highly aligned nanofibers of poly(l-lactic acid) (PLLA) through an electrospinning technique, followed by surface polymerization with a conductive monomer of 3,4-ethoxylene dioxythiophene (EDOT), and evaluated their effects on MSCs under electric fields. Notably, better cell adhesion and proliferation were observed on the surface of fabricated nanofibers [114]. Impact of other scaffolds on stem cell functions is summarized in Table 3. 

It is well established that the scaffold materials closely resemble the native ECM and are frequently used in tissue engineering applications for replacing damaged or diseased tissues. Wang and coworkers developed the 3D-printed porous scaffolds from a poly(propylene fumarate) (PPF) resin and evaluated the cytotoxicity of the degraded scaffolds through (XTT assay). XTT (sodium 3′-[1-[(phenylamino)-carbonyl]-3, 4-tetrazolium]-bis (4-methoxy-6-nitro) benzene sulfonic acid hydrate is a tetrazolium salt, which form water-soluble formazan on bioreduction. Degradation is an important parameter of the polymer scaffolds, which indicates their durability. PPF has ester groups in its structure and can be hydrolyzed into fumaric acid and propylene glycol. It was seen that over a 224-day period, the 3D-printed polymer scaffolds were hydrolytically degraded and changes in their physical appearance, such as porosity and pore size, occurred. No significant decrease in fibroblast cell viability was observed in the presence of degraded scaffolds, indicating their biocompatibility. This result suggests that degradable, 3D-printed PPF scaffolds are an ideal material for tissue engineering, especially for bone tissues [141]. The effect of nanohydroxyapatite, antibiotic, and mucosal defensive agents on the mechanical and thermal properties of glass ionomer cement was evaluated by Chieruzzi et al. for clinical purposes. They noted the mechanical and thermal responses were deeply affected by the filler content [142]. Sometimes, inflammation has also been noted near the implantation site in the host [143]. Therefore, it is necessary to design materials that have the potential to regulate the host immune response directly [144]. Furthermore, many ECM has immunomodulatory domains that directly bind to the immune cells and regulate their functions [145]. An immunomodulatory response of naturally derived ECM materials is summarized in Table 4. A schematic representation of interactions between the ECM and immune cells is shown in Figure 6**.**


## 4. Conclusions

Several studies have been conducted to investigate the impact of scaffold materials on stem cell function. Synthetic as well as natural polymer scaffolds provide suitable microenvironments to the cells, enabling better attachment, proliferation, and differentiation. Modifications of natural or synthetic polymer scaffolds are required to meet the desired criteria of an ECM for enhanced cellular responses. Notably, better cell attachment, proliferation, and differentiation were observed on rough surfaces of scaffolds compared to smooth surfaces. In this review, we have described different kinds of scaffolds and their impact on stem cell activities. We hope that scaffold materials will become very practical and attractive tools for regenerative tissue engineering.

## Figures and Tables

**Figure 1 materials-12-02950-f001:**
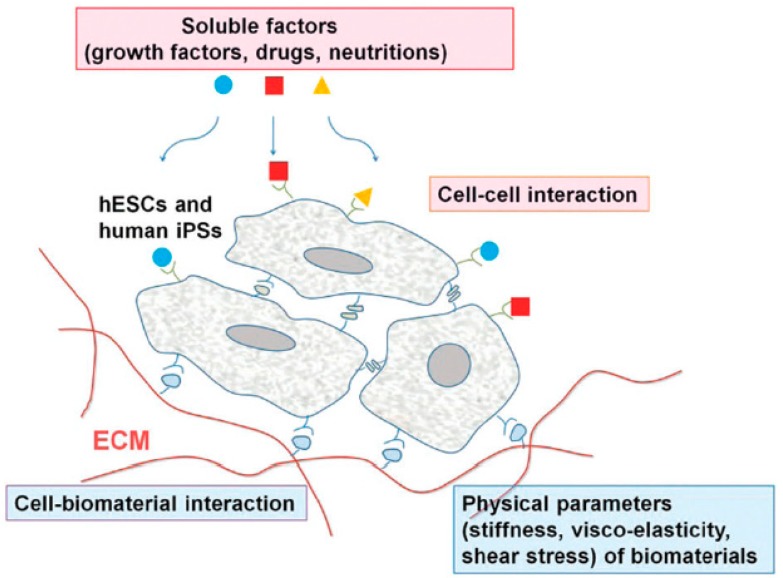
Schematic representation of the microenvironments and niches of stem cells and their regulation by the following factors: soluble factors, such as growth factors or cytokines, nutrients, and bioactive molecules; cell–cell interactions; and cell–biomaterial interactions. Physical properties of biomaterials also regulate the stem cell fate. Reproduced with permission from reference [5]; Copyright 2015, Royal Society of Chemistry.

**Figure 2 materials-12-02950-f002:**
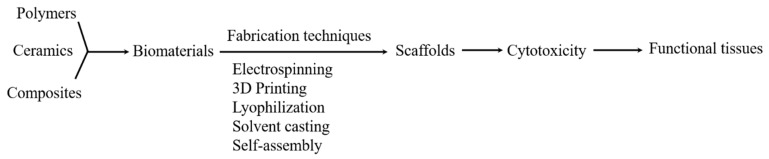
A flowchart for the generation of the functional tissues from biomaterials (from reference [31]).

**Figure 3 materials-12-02950-f003:**
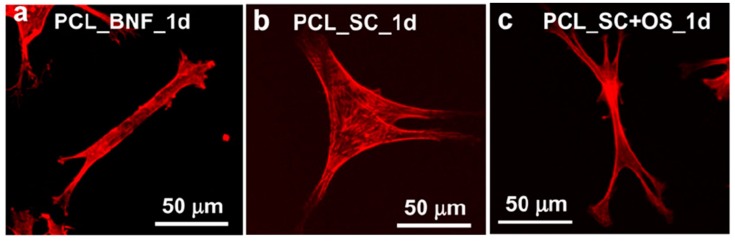
(**a**–**c**) Effect of different types of scaffolds on the morphology of human bone marrow stromal cells (hBMSC) (400×) after 1 day of culturing. (**a**) PCL_BNF: big nanofiber scaffolds ~900 nm in diameter, (**b**) PCL_SC: thin film of PCL polymer produced through spin coating, and (**c**) PCL_SC+OS: thin film with osteogenic supplements, and 1_d: 1 day. Reproduced with permission from references [92]; Copyright 2011, Elsevier.

**Figure 4 materials-12-02950-f004:**
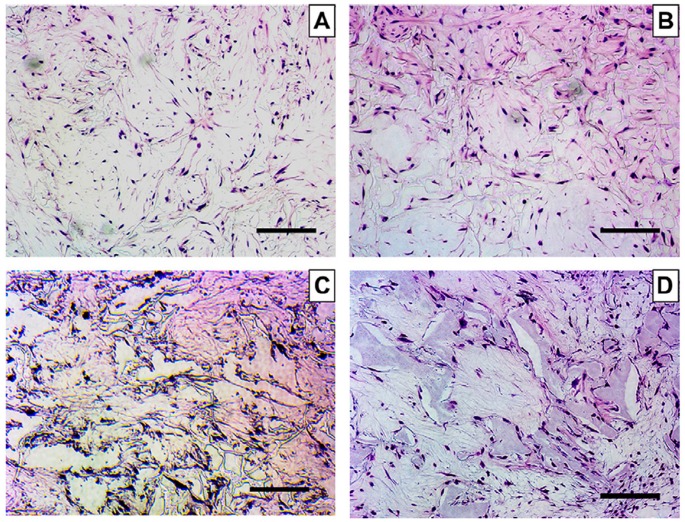
Hematoxylin and Eosin (H&E) staining of human dental pulp stem cells (DPSCs) cultured on solid-walled (SW) and nanofibrous (NF) poly(l-lactic acid) (PLLA) scaffolds in odontogenic media: (**A**) on SW-PLLA scaffolds for 4 weeks; (**B**) on NF-PLLA scaffolds for 4 weeks; (**C**) on SW-PLLA scaffolds for 8 weeks; (**D**) on NF-PLLA scaffolds for 8 weeks. Scale bars represent 100 mm. Reproduced with permission from reference [101]; Copyright 2011, Elsevier.

**Figure 5 materials-12-02950-f005:**
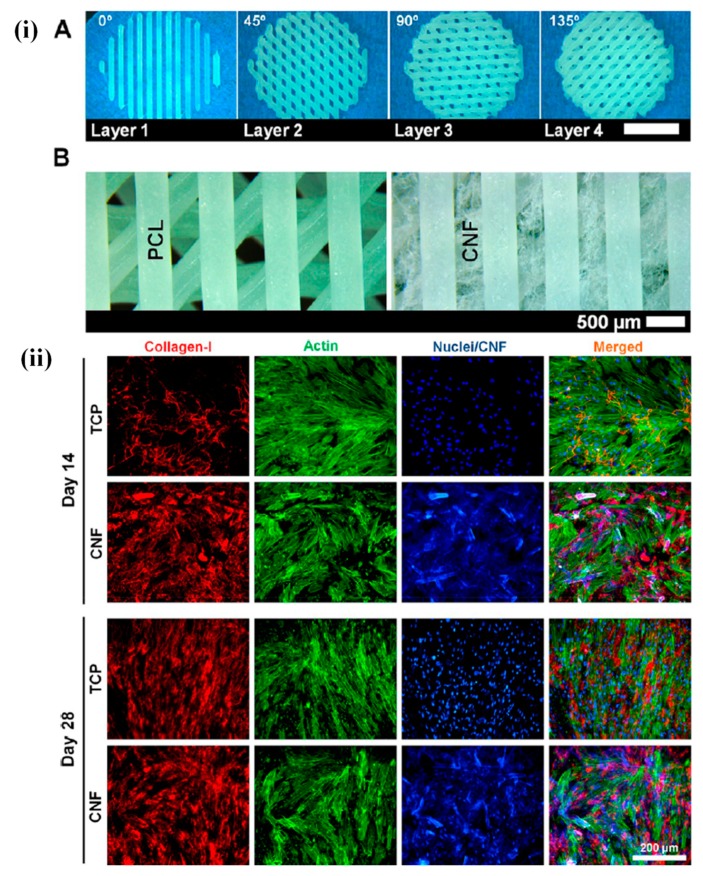
(**i**) Structural characterization of the 3D-printed PCL scaffolds. (**A**) Stereomicroscope images of the printed four layers with orientations of 0°–45°–90°–135° and strand spacing of 1 mm (scale bar: 5 mm). (**B**) Stereomicroscope images of the 3D-printed PCL before and after coating with CNF, showing the cellulose fibers between the PCL strands. (**ii**) collagen type I production and actin cytoskeleton organization of cells after 14 and 21 days. Reproduced with permission from reference [109]; Copyright 2018, American Chemical Society.

**Figure 6 materials-12-02950-f006:**
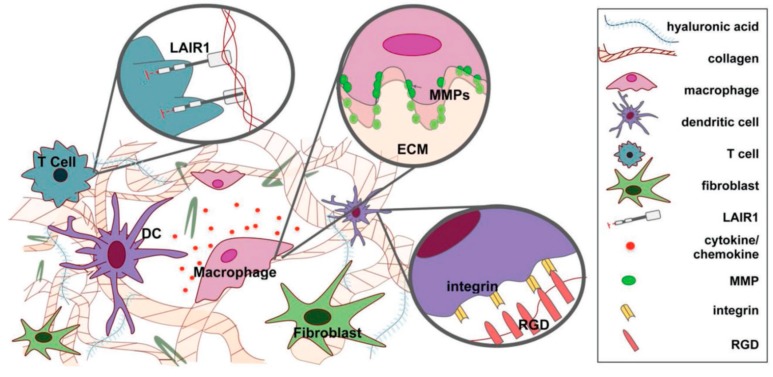
Schematic representation of ECM–immune cell interactions. Interactions include the LAIR1–collagen interaction that inhibits inflammatory signaling, matrix metalloproteinases (MMPs) that drive matrix degradation at cleavage motifs, and RGD (Arginylglycylaspartic acid) that facilitates cellular adhesion to ECM via integrin binding [150]; Copyright 2019, John Wiley and Sons.

**Table 1 materials-12-02950-t001:** Some common synthetic and natural polymers used for hydrogel synthesis. Reproduced with permission from reference [49]; Copyright 2012, Elsevier.

Natural Polymers and Their Derivatives (±Crosslinkers)				Synthetic Polymers (±Crosslinker) Polyesters	Other Polymers	Combinations of Natural and Synthetic Polymers
Anionic polymers: Hyaluronic acid (HA), alginic acid, pectin, carrageenan, chondroitin sulfate, dextran sulfate	Cationic polymers: Chitosan, polylysine	Amphipathic polymers: Collagen (and gelatin), carboxymethyl chitin, fibrin	Neutral polymers: Dextran, agarose, pullulan	poly(ethylene glycol)–poly(lactic acid)–poly(ethylene glycol) (PEG–PLA–PEG), PEG–poly(lactic-*co*-glycolic acid) (PLGA)–PEG, PEG–polycaprolactone (PCL) –PEG, PLA–PEG–PLA, poly(hydroxy butyrate) (PHB), poly(propylene fumarate)-*co*-(ethylene glycol) (P(PF-*co*-EG)) ± acrylate end groups, PEG–poly(butylene oxide) (PBO) terephthalate	PEG–bis-(PLA-acrylate), PEG ± cyclodextrin (CD), PEG–g-poly(acrylamide(AAm)-*co*-Vamine), poly(N-isopropylacrylamide-*co*-acrylic acid) P(NIPAAm-*co*-AAc), P(NIPAAm-*co*-ethyl methacrylate (EMA), poly(vinyl acetate)–poly(vinyl alcohol) (PVAc–PVA), poly(N-vinyl pyrrolidone) (PNVP), poly(methyl methacrylate-*co*-hydroxyethyl methacrylate) (PMMA-*co*-MEHA), poly(acrylonitrile-*co*-allyl sulfonate), poly(biscarboxy-phenoxy-phosphazene), poly(glucosylethylmethacrylate) P(GEMA-sulfate)	P(PEG-*co*-peptides), alginate–g-poly(ethylene oxide)–poly(propylene oxide)–poly(ethylene oxide) (alginate–g-PEO–PPO–PEO), P(PLGA-*co*-serine), collagen acrylate, alginate acrylate, poly(hydroxypropyl methacrylamide-g-peptide) P(HPMA-g-peptide), poly(hydroxyethyl methacrylate-Matrigel^®^) P(HEMA-Matrigel^®^), HA-g-NIPAAm

**Table 2 materials-12-02950-t002:** Common scaffold fabrication techniques and their applications.

Method/Scaffold	Polymers Used	Applications	References
**For gel scaffold fabrication**			
Emulsification technique	Collagen, gelatin, and hyaluronic acid (HA)	Controlled drug delivery	[50,66,67]
Micromolding process	Poly(ethylene glycol) (PEG), HA, alginate, poly(methyl methacrylate) (PMMA)	Delivery of small-molecule-like drugs and insulin	[68,69,70]
Microfluidics process	Calcium alginate, PEG, silicon, poly(dimethyl siloxane) (PDMS)	Sensing, cell separation, and controlled microreactors	[71,72,73]
Photolithography technique	Chitosan, PMMA, PEG, Poly(2-(trimethylamino)ethyl methacrylate (PDMAEM)	Cell–cell interactions, biosensors, microdevices	[74,75,76]
Injectable gel scaffold	Copolymers of poly(lactic acid) (PLA), poly(glycolic acid) (PGA), PEG, poly(lactic-*co*-glycolic acid) (PLGA), copolymers of poly(ethylene oxide) (PEO), chitosan, collagen, and HA	Cartilage and bone tissue engineering, drug delivery	[77,78,79]
**For porous scaffold fabrication**			
Solvent casting/salt leaching technique	Collagen, PLGA, poly(l-lactic acid) (PLLA)	Cartilage and bone tissue engineering,	[80,81,82]
Gas foaming/salt leaching technique	PLLA, PLGA, poly(d,l-lactic acid) (PDLLA)	Delivery of small molecules such as drugs, tissue engineering	[83,84,85]
Ice particle leaching technique	PLLA and PLGA	Bone tissue engineering	[86,87,88]

**Table 3 materials-12-02950-t003:** Impact of other scaffolds materials on stem cell fate.

Materials	Nature of Scaffold	Impact on Different Stem Cells	References
Calcium phosphate–chitosan composite Chitosan	Injectable scaffolds Fibrous scaffolds	Cell proliferation and osteogenic differentiation Chondrogenesis	[115,116]
Poly(caprolactone) (PCL) Poly(l-lactic acid)-*co*-poly(3-caprolactone)/collagen PCL/hydroxyapatite PCL PCL/polydopamine	Freeform fabricated (FFF) scaffolds Nanofibrous scaffolds Coiled scaffolds Microfibrous scaffolds	Cell proliferation and differentiation Hepatic trans-differentiation Osteogenesis Osteogenesis Mild myofibroblastic differentiation	[96,117,118,119,120]
Polyethylene oxide and poly(3-hydroxybutyrate-*co*-3-hydroxyvalerate)	Nanofibrous scaffolds	Neuronal differentiation and peripheral nerve regeneration	[121]
Cartilage-derived Collagen/gold-coated collagen Starch Fibrin Nanostructured tendon-derived biomaterials	Crosslinked scaffolds Crosslinked scaffolds Nanofibers 3D scaffolds Nanofibrous scaffolds	Chondrogenesis Chondrogenesis and osteogenesis Enhanced differentiation and proliferation Osteogenesis Neuronal differentiation Enhanced osteogenesis	[122,123,124,125,126,127]
Poly(lactic acid)/silk fibroin	Nanofibrous scaffolds	Neuronal differentiation	[128]
Gelatin methacrylate Gelatin Hyaluronic acid (HA) Poly(ethylene glycol) (PEG)	Hydrogels	Neuronal differentiation Enhanced osteogenesis Cell differentiation	[129,130,131,132]
Graphene foam Polyurethane foam	3D porous structure	Enhanced neuronal differentiation Promoted hepatogenesis	[133,134]
Conducting polymer (CP)-based biomaterials	Thin film/nanofibers/scaffolds	Enhanced osteogenesis	[135,136,137,138,139,140]

**Table 4 materials-12-02950-t004:** Immunomodulatory impact of naturally derived ECM.

Materials	Immunomodulatory Effect of the ECM	References
Collagen/chemically modified collagen/denatured collagen (gelatin)	Degranulation of peripheral basophils and suppressed immune cell activity/lower inflammatory response/anti-inflammatory response	[146,147,148,149]
Fibrin-based materials	Inflammatory anti-inflammatory effects	[150]
Hyaluronic acid (HA)	Dependent upon the molecular weight (MW) of HA; high-MW HG was shown to be inert or immunosuppressive, and lower-MW HA provoked the inflammatory response	[151,152,153,154]
Decellularized matrices	Anti-inflammatory	[155,156,157,158,159]
Engineered ECM peptide-mimetic materials	Both anti-inflammatory and inflammatory responses	[160,161,162,163,164,165,166]

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
