# Peer review of "Biomimetic Polymer-Based Engineered Scaffolds for Improved Stem Cell Function"

_materials, 2019, doi:10.3390/ma12182950_

Round 1

Reviewer 1 Report

The manuscript entitled “Biomimetic polymer-based engineered scaffolds for improved stem cell functions” describes the major classes of biodegradable polymer scaffolds and their impact on stem cell functions. This revision manuscript focuses a very interesting subject and present adequate references. However it presents some weaknesses mainly related with English aspects, it gives a too succinct description of important concepts, it is repetitive and not so accurate sometimes.

Consequently, corrections are needed and authors should present the information in a more clear, structured and precise way.

Some detailed points which should be clarified:

- The English should be revised. (Some examples of corrections needed although there is much more: lines 28, 67, 73, 96, 147, 256, 317,...)

- The manuscript focuses on biomimetic polymer-based scaffolds but the term biomimetic is just used once in the entire manuscript almost at the end of it (line 255). Please pay attention to this when reformulate the manuscript. Important terms/definitions should be addressed in more detail.

- Despite presenting an adequate and considerable amount of references the work is too succinct. Being a revision work a more extensive, comprehensive and accurate description of some concepts should be presented, like pluripotency for instance (line 29), definition of hydrogels scaffolds (from line 96), etc…

- Authors should also be aware that even with the same composition, scaffolds obtained by different techniques may/will present different properties and thus induce different cells responses. This is very important and is not clear in the text.

- Line 18: replace “were changed” by “are different” or “can be different”

- Line 31, explain better why “itself”.

- Lines 54 and 130, authors mention the use of “fillers” to tune scaffolds properties, however, this is not scientifically correct. It can be used other polymers, additives, etc, that are not necessarily  fillers - please correct.

- Lines 79-94: saying that the major advantage of 3D printing is the reduction of waste material is very reductive and just correct in some particular cases. Need to be reformulate. Also, at the scope of the manuscript I do not think it makes much sense to mention aerospace applications, better to focus only at biomedical applications.

- Figure 2 legend, it should be said which is the composition.

- Legend of tables should appear at the top of them.

- Tables should be revised.

Author Response

Reviewer 1:

Comments and Suggestions for Authors

The manuscript entitled “Biomimetic polymer-based engineered scaffolds for improved stem cell functions” describes the major classes of biodegradable polymer scaffolds and their impact on stem cell functions. This revision manuscript focuses on a very interesting subject and presents adequate references. However it presents some weaknesses mainly related to English aspects, it gives a too succinct description of important concepts, it is repetitive and not so accurate sometimes.

Consequently, corrections are needed and authors should present the information in a more clear, structured and precise way.

Reply to Reviewer: We appreciate the reviewer positive recommendations for the manuscript and their valuable comments for future improving the paper. Some detailed points which should be clarified:

- English should be revised. (Some examples of corrections needed although there is much more: lines 28, 67, 73, 96, 147, 256, 317,...)

Response: Thank you very much for your kind suggestion. We have revised the manuscript to improve the English as per your valuable suggestions.

- The manuscript focuses on biomimetic polymer-based scaffolds but the term biomimetic is just used once in the entire manuscript almost at the end of it (line 255). Please pay attention to this when reformulating the manuscript. Important terms/definitions should be addressed in more detail.

Response: Thank you very much for your kind suggestion. We have considered more attention to the keywords like biomimetic in the revised manuscript.

- Despite presenting an adequate and considerable amount of references the work is too succinct. Being a revision work a more extensive, comprehensive and accurate description of some concepts should be presented, like pluripotency for instance (line 29), the definition of hydrogels scaffolds (from line 96), etc.

Response: Thank you very much for reviewing the manuscript. We have revised the manuscript and added more description of some concept like pluripotency and hydrogel scaffolds.

- Authors should also be aware that even with the same composition, scaffolds obtained by different techniques may/will present different properties and thus induce different cells responses. This is very important and is not clear in the text.

Response: Thank you very much for reviewing the manuscript. We agree with your point of view. We have briefly described it in the manuscript.

- Line 18: replace “were changed” by “are different” or “can be different”

Response: Thank you very much for your kind suggestion. We have replaced the word “were changed” by “are different” in the manuscript.   

- Line 31, explain better why “itself”.

Response: Thank you very much for reviewing the manuscript. Here, the word itself indicates the proper differentiation and the maintenance of pluripotency are regulated not only by the surrounding microenvironments but also stem cells themselves. 

- Lines 54 and 130, authors mention the use of “fillers” to tune scaffolds properties, however, this is not scientifically correct. It can be used other polymers, additives, etc, that are not necessarily fillers - please correct.

Response: Thank you very much for your kind suggestion. We have added the additives, as well as polymer words with the filler in the manuscript.

- Lines 79-94: saying that the major advantage of 3D printing is the reduction of waste material is very reductive and just correct in some particular cases. Need to be reformulated. Also, at the scope of the manuscript, I do not think it makes much sense to mention aerospace applications, better to focus only at biomedical applications.

Response: Thank you very much for reviewing the manuscript. We agree with you and reformulated the sentence. In this manuscript, we are mainly focused on the biomedical applications of different scaffolds. Here, we have briefly mentioned another possible application of the 3D printing technique.

- Figure 2 legend, it should be said which is the composition.

Response: Thank you very much for reviewing the manuscript. We have mentioned the composition of the fiber in the Figure 2 legend.

- Legend of tables should appear at the top of them.

Response: Thank you very much for reviewing the manuscript. We have rearranged the table legend as per your suggestion.

- Tables should be revised.

Response: Thank you very much for reviewing the manuscript. We have revised the tables.

Reviewer 2 Report

this is an interesting narrative review about scaffolds use and properties in engineering use.
Some criticism are present:

.Introduction section: Some important considerations must be added to introduction section about potential scaffold use in medicine, surgery, dentistry and other applications. About this aspect the following article must be added and discussed about, for example in dentistry:

Chieruzzi M, Pagano S, Moretti S, Pinna R, Milia E, Torre L, Eramo S Nanomaterials for Tissue Engineering in dentistry. Nanomaterials. 2016 Jul 21;6(7)

-Even if is a descrittive review, some information about the flowchart of the work must be added. No informations can be desumed from the section about inclusion and exclusion criterias, or keyword or research strategies-

Line 77 A table 1 must be added to insert all scaffold type and production (3d, Hydrogel scaffold, etc)

-Table 1 must be rewritten, it is not correct inser an other articles table in the text

Line 125 The author must be add references about this aspect

Some consideration must be added, in Discussion section, about cytotoxicity present in literatura about scaffold, techniques and implications.

About this aspect i suggest to add the following article in reference section:

Chieruzzi M, Pagano S, Lombardo G, Marinucci L, Kenny JM, Torre L, Cianetti S Effect of nanohydroxyapatite antibiotic, and mucosal defensive agent on the mechanical and thermal properties of glass ionomer cements for special needs patients Journal of Materials research 2018,33(6):638-649

Author Response

Reviewer 2:

Comments and Suggestions for Authors:

This is an interesting narrative review of scaffolds use and properties in engineering use.
Some criticism is present:

Reply to Reviewer: We appreciate the reviewer positive recommendations for the manuscript and their valuable comments for future improving the paper.

Introduction section: Some important considerations must be added to the introduction section about potential scaffold use in medicine, surgery, dentistry and other applications. About this aspect the following article must be added and discussed, for example in dentistry:

Chieruzzi M, Pagano S, Moretti S, Pinna R, Milia E, Torre L, Eramo S Nanomaterials for Tissue Engineering in dentistry. Nanomaterials. 2016 Jul 21;6(7)

Response: Thank you very much for your kind suggestion. We have added and discussed other applications of the scaffolds with citation of the given article in the introduction section of the manuscript.

-Even if is a descriptive review, some information about the flowchart of the work must be added. No information can be resumed from the section about inclusion and exclusion criteria, or keyword or research strategies.

Response: Thank you very much for your kind suggestion. We have added the flowchart in the manuscript.

Line 77 table 1 must be added to insert all scaffold type and production (3d, Hydrogel scaffold, etc.)

Response: Thank you very much for reviewing the manuscript. Table 1 is focused on only hydrogel scaffolds. We have given the other scaffold materials (3D, Fibrous) in Table 3.

Table 1 must be rewritten, it is not correct to insert another articles table in the text.

Response: Thank you very much for reviewing the manuscript. We have re-written the Table 1.

Line 125 the author must add references to this aspect.

Response: Thank you very much for reviewing the manuscript. We have added a reference there.

Some consideration must be added, in the Discussion section, about cytotoxicity present in literature about scaffold, techniques, and implications. About this aspect I suggest to add the following article in the reference section:

Chieruzzi M, Pagano S, Lombardo G, Marinucci L, Kenny JM, Torre L, Cianetti S Effect of nanohydroxyapatite antibiotic, and mucosal defensive agent on the mechanical and thermal properties of glass ionomer cement for special needs patients Journal of Materials Research 2018,33(6):638-649.

Response: Thank you very much for reviewing the manuscript. We have briefly described the cytotoxicity of the 3D-printed polymer scaffolds and added the mentioned reference in the manuscript.

Round 2

Reviewer 1 Report

Authors have accepted the reviewer recommendations/suggestions and manuscript is now acceptable for publication. A minor revision is still needed:

Line 9: … properties instead of property

Line 96: researchers instead of researcher

Line 125: PU and PLGA– first time that these abbreviations appear on the manuscript it should be written the full name

Line 134: PDLLA – the same as mentioned above

Author Response

Comments and Suggestions for Authors

Authors have accepted the reviewer recommendations/suggestions and manuscript is now acceptable for publication. A minor revision is still needed:

Reply to Reviewer: We appreciate the reviewer positive recommendations for the manuscript and their valuable comments for future improving the paper.

Line 9: … properties instead of property

Response: Thank you very much for your kind suggestion. We have added the correct word.

Line 96: researchers instead of a researcher

Response: Thank you very much for your in-depth reviewing of the manuscript. We have added the correct word.

Line 125: PU and PLGA– the first time that these abbreviations appear on the manuscript it should be written the full name

Response: Thank you very much for reviewing the manuscript. We have described the full name of PU and PLGA in the manuscript.

Line 134: PDLLA – the same as mentioned above

Response: Thank you very much for your kind suggestion. We have mentioned the full name of PDLLA in the revised manuscript.
